# The Novel Class IIa Selective Histone Deacetylase Inhibitor YAK540 Is Synergistic with Bortezomib in Leukemia Cell Lines

**DOI:** 10.3390/ijms232113398

**Published:** 2022-11-02

**Authors:** Lukas M. Bollmann, Alexander J. Skerhut, Yodita Asfaha, Nadine Horstick, Helmut Hanenberg, Alexandra Hamacher, Thomas Kurz, Matthias U. Kassack

**Affiliations:** 1Institute of Pharmaceutical and Medicinal Chemistry, Heinrich-Heine-University Duesseldorf, 40225 Duesseldorf, Germanythomas.kurz@hhu.de (T.K.); 2Department of Otorhinolaryngology and Head/Neck Surgery, Heinrich Heine University, 40225 Duesseldorf, Germany

**Keywords:** leukemia, acute myeloid leukemia, histone deacetylase (HDAC) inhibition, proteasome inhibitor, bortezomib, YAK540, TMP269, class IIa HDAC inhibition, epigenetics

## Abstract

The treatment of leukemias, especially acute myeloid leukemia (AML), is still a challenge as can be seen by poor 5-year survival of AML. Therefore, new therapeutic approaches are needed to increase the treatment success. Epigenetic aberrations play a role in pathogenesis and resistance of leukemia. Histone deacetylase (HDAC) inhibitors (HDACIs) can normalize epigenetic disbalance by affecting gene expression. In order to decrease side effects of so far mainly used pan-HDACIs, this paper introduces the novel highly selective class IIa HDACI YAK540. A synergistic cytotoxic effect was observed between YAK540 and the proteasome inhibitor bortezomib (BTZ) as analyzed by the Chou-Talalay method. The combination of YAK540 and BTZ showed generally increased proapoptotic gene expression, increased *p21* expression, and synergistic, caspase 3/7-mediated apoptosis. Notably, the cytotoxicity of YAK540 is much lower than that of pan-HDACIs. Further, combinations of YAK540 and BTZ are clearly less toxic in non-cancer HEK293 compared to HL-60 leukemia cells. Thus, the synergistic combination of class IIa selective HDACIs such as YAK540 and proteasome inhibitors represents a promising approach against leukemias to increase the anticancer effect and to reduce the general toxicity of HDACIs.

## 1. Introduction

Leukemia is the summary of various forms of blood cancer. The four main forms of leukemia are acute lymphoblastic leukemia (ALL), acute myeloid leukemia (AML), chronic lymphocytic leukemia (CLL) and chronic myeloid leukemia (CML). The pooled 5-year survival rate from all leukemias is 65.7% over 2012–2018 in the United States [1]. This demonstrates that leukemias can be treated reasonably well, although a more nuanced view shows that, for example, the 5-year survival rate differs greatly between acute myeloid leukemia and chronic myeloid leukemia. For example, the 5-year survival rate for acute myeloid leukemia is only 30.5% between 2012 and 2018 in the United States, whereas the 5-year survival rate for chronic myeloid leukemia is 70.4% over the same period [2,3]. The unmet medical need to improve AML treatment is also reflected in the recorded deaths from AML (51689) compared to 5653 in chronic myeloid leukemia between 2014 and 2018 in the United States, and furthermore, the incidence of AML along with chronic lymphatic leukemia, is among the leaders in the field [4]. For this reason, it is important to develop new and even better therapeutic concepts to enable more successful treatment of leukemias, especially AML.

In the past, proteasome inhibitors, such as bortezomib (BTZ), have been shown to be a promising approach for the treatment of hematological disorders. Proteasome inhibitors are approved for the treatment of multiple myeloma [5]. Bortezomib-induced inhibition of the proteasome leads to the accumulation of ubiquitinylated, damaged proteins affecting various cellular processes such as cell cycle, DNA repair, transcription, cell signaling pathways, cell survival and apoptosis [6]. Due to the high proliferation rate and increased protein synthesis rate, malignant cells are more sensitive to inhibition of the proteasome [7]. The use of BTZ is however limited by severe side effects such as neuropathy and hemotoxicity [8,9]. Therefore, a potential dose reduction of BTZ to minimize its side effects or, alternatively, less toxic proteasome inhibitors are highly desirable. A recent study investigated the proteasome inhibitor ixazomib with an improved safety profile, such as reduction of peripheral neuropathy, and BTZ on ALL and AML cell lines [10]. BTZ is more potent than ixazomib, but this may be compensated by an improved pharmacokinetic and toxicological profile of ixazomib [10,11,12,13]. Several studies investigated BTZ in combination with other cytotoxic agents against AML from the groups of anthracyclines (e.g., doxorubicin), DNA methyltransferase inhibitors (e.g., azacitidine), and histone deacetylase inhibitors (HDACI) such as vorinostat [6]. In addition, carfilzomib, another proteasome inhibitor, showed superadditive cytotoxic effect on multiple myeloma cells in vitro in combination with the class IIa HDACI TMP269 [14].

Through epigenetic modulation, histone deacetylases (HDACs) have an impact on various cellular processes [15]. HDACs can be divided into four classes. Class I includes HDAC 1, 2, 3, and 8, which are present in the nucleus. Class II can be divided into IIa (HDAC 4, 5, 7 and 9) and IIb (HDAC 6 and 10). These HDACs are present in both the nucleus and cytoplasm [16]. Class III do not contain zinc dependent HDACs. Class IV contains HDAC 11, which has properties of class I and II [15]. Dysregulation of HDACs play a role in the development of certain cancers by altering transcription. The use of HDACIs is intended to restore the dysregulation of HDACs in cancer cells to normal. By regulating pro- and anti-apoptotic genes, HDACIs can induce apoptosis [15]. Another point is that the combination treatment of HDACIs with, for example, bortezomib could increase the sensitivity of cancer cells, so that a dose reduction of bortezomib would be possible.

To minimize the potential side effects of BTZ by dose reduction on the one hand and to reduce the potential side effects of unselective (pan) HDACIs on the other hand, the use of selective class IIa inhibitors in combination with the proteasome inhibitor was tested. One of the first potent class IIa HDACIs is TMP269 which has however the disadvantage of poor solubility [17]. We intended to improve the physicochemical properties and potency of TMP269 and thus developed YAK540 as a new selective class IIa HDACI. In the meantime, further improved class IIa selective HDACIs were developed such as NT160 [18] and compound 12 [19]. Yet, in this study we compare the first well-established trifluoromethyloxadiazolyl (TFMO)-type class IIa selective inhibitor TMP269 with our novel inhibitor YAK540. Both, YAK540 and TMP269 were evaluated for a synergistic effect in combination with BTZ in various leukemia cell lines with a focus on AML.

The pharmacophore model for class IIa selective HDACI contains the following four elements: a zinc binding group (ZBG), a linker that interacts with the substrate binding tunnel, a lower pocket group (LP-group) that occupies the selectivity pocket and a cap, also known as surface recognition domain (Figure 1) [20]. In 2013, Lobera et al. and Novartis discovered simultaneously class IIa selective HDACIs, revealing a trifluoromethyloxadiazolyl moiety (TFMO) as a novel ZBG. Lobera et al. demonstrated that inhibitors exhibiting this novel ZBG showed up to 150–10,000 fold higher selectivity towards class IIa HDACs compared to their corresponding hydroxamates [21]. A crystallographic analysis confirmed that the TFMO group acts as a non-chelating ZBG via the coordination of the fluorine atoms and its oxygen to the zinc ion in the catalytic center. It is proposed that class IIa selectivity arises from the bulkiness but modest zinc binding ability of the TFMO moiety as well as the U-shaped conformation of these inhibitors. Lobera et al. reported that the TFMO-based HDACI display an improved pharmacokinetic profile compared to hydroxamates [21]. Furthermore, it is presumed that the TFMO series has fewer pan inhibitor associated off-targets effects.

## 2. Results

### 2.1. Synthesis of YAK540

The first building block was the respective hydroxylamine **3**. 1-(2-Chloroethyl)piperidine hydrochloride (**1**) was converted with N-hydroxyphthalimide (NHPI) to generate **2**. Subsequently, the deprotection was performed with 2.00 eq of hydrazine monohydrate yielding the desired hydroxylamine **3** (Figure 1).

YAK540 was generated, via a HATU-mediated coupling of the respective acid **4** and the hydroxylamine **3** in moderate yield (Figure 2, Appendix A).

### 2.2. Investigation of Cytotoxicity and HDAC Inhibitory Effect

First, the cellular HDAC inhibition activity and cell permeability of YAK540 and TMP269 were determined. For this purpose, cellular HDAC assays were performed using a class IIa HDAC and HDAC 8 specific substrate Boc-Lys-(TFa)-AMC in addition to a class I and IIb HDAC substrate Boc-Lys (Ac)-AMC [22,23]. This allowed differentiating the cellular activity of class IIa HDACs from the remaining HDACs. The results for YAK540 and TMP269 are shown in Figure 2.

The IC_50_ value of YAK540 using the class IIa selective substrate Boc-Lys-(TFa)-AMC was 590 nM whereas the IC_50_ value using the class I/IIb substrate Boc-Lys-(Ac)-AMC was 64.2 µM. This result showed a high (factor 108) cellular selectivity of YAK540 for class IIa HDACs. In contrast to YAK540, TMP269 appeared with a much lower cellular class IIa HDAC inhibition (IC_50_ value around 100 µM) and no class I/IIb HDAC inhibition up to 100 µM.

Next, YAK540 and TMP269 were investigated in HDAC enzyme assays using HDAC 2, HDAC 4, HDAC 6, and HDAC 8 as representatives for their respective HDAC classes (Table 1).

YAK540 showed excellent potency at HDAC 4 as the major representative of class IIa HDACs with an IC_50_ of 0.114 µM. Further, YAK540 indicated excellent selectivity for HDAC 4 against HDAC 2 (265-fold), against HDAC 6 (100-fold), and against HDAC 8 (82-fold). Thus, YAK540 presented at least 82-fold selectivity for HDAC 4 compared to the other enzymes. Compared to TMP269, YAK540 had improved selectivity. TMP269 had more than 600-fold selectivity for HDAC 4 compared to HDAC 2, but only 27-fold selectivity for HDAC 4 compared to HDAC 8, based on literature data for enzyme inhibition [22]. Another problem related to TMP269 is its poor solubility. We observed precipitations at higher concentrations of TMP269 in aqueous assay buffer, which was not the case for YAK540, requiring to add further solubilizers such as DMSO to TMP269 dilution series. YAK540 thus has improved aqueous solubility which clearly is advantageous over TMP269.

We next determined the cytotoxic effects of YAK540, TMP269, and BTZ on four different leukemia cell lines, namely the AML cell lines HL-60, THP-1, and MONO-MAC-6, and the CML cell line K562 (Figure 3). BTZ was expectedly cytotoxic in the double-digit nanomolar range. Interestingly, YAK540 had a significantly lower cytotoxic activity compared to TMP269. The IC_50_ value of TMP269 was around 20 µM in all leukemia cell lines examined, whereas the IC_50_ of YAK540 was predominantly above 100 µM. The IC_50_ values determined are listed in Table 2.

### 2.3. Apoptosis Induction and Caspase Activation of Class IIa Inhibitors in Combination with Proteasominhibitor Bortezomib

Next, we investigated combinations of BTZ and YAK540 or TMP269 and analyzed if the cytotoxic effect is based on caspase-mediated apoptosis. SubG1 analysis of BTZ, YAK540 and TMP269 alone and in combinations in THP-1 cells is illustrated in Figure 4. Whereas 5 µM TMP269 or 5.7 µM YAK540 showed only very little subG1 induction, combinations of TMP269 or YAK540 with BTZ demonstrated over additive effects for both concentrations of BTZ used in this experiment (Figure 4a). Figure 4b displays the difference between an additive effect only (black bars) where the effects of single treatments with BTZ and YAK540 or TMP269 were simply added compared to the effect of the combination of BTZ with YAK540 or TMP269 (white bars). Incubation of the combination leads to a superadditive (=synergistic) effect.

In the next step, we investigated whether apoptosis is induced in a caspase-dependent manner. For this purpose, caspase 3/7 activation was measured by fluorescence imaging of YAK540, TMP269, and BTZ treated THP-1 cells (Figure 5; Appendix A). Whereas YAK540, TMP269, and 3.8 nM BTZ alone showed (almost) no caspase activation, 7.9 nM BTZ indicated clear caspase activation and combination of 3.8 nM BTZ with YAK540 but not TMP269 increased caspase activation. To analyze over-additive effects, single effects of the compounds were added (white bars) and compared to combination effects (YAK540 plus BTZ; TMP269 plus BTZ). Whereas the combinations of 5.7 µM YAK540 plus BTZ illustrated over-additive (=synergistic) effects for both BTZ concentrations, only the combination of 5 µM TMP269 with 7.9 nM BTZ showed a synergistic effect. Fluorescence images showing nuclear staining (Hoechst 33342) and cleaved fluorescent caspase substrate are available in the Appendix A).

### 2.4. Chou-Talalay Synergism Studies in MTT Assays

To determine whether a synergistic effect exists, Chou-Talalay combination studies were performed using MTT assay [24]. The combination index (CI) was calculated for different concentrations of BTZ with the class IIa HDACIsYAK540 and TMP269 (Table 3). Cells were co-incubated with BTZ and the HDACI. A synergistic effect is indicated by a CI value of less than 0.9. All cell lines exhibited CI values smaller than 0.9, and in some cases smaller than 0.5, indicating strong synergism. In addition, the synergy levels of the combination-concentration-response surface were graphically mapped by Combenefit. For calculating the synergy levels, the Bliss model was used. (Figure 5) [25,26].

As can be seen in the combenefit graphs based on the Bliss model in Figure 6, HL-60 and MONO-MAC-6 show significantly stronger synergy with YAK540 plus BTZ than TMP269 plus BTZ.

### 2.5. Mechanistic Study of Class IIa Inhibitors in Combination with Proteasome Inhibitors

The next step was the detailed mechanistic investigation of the combination therapy. For this purpose, RT-qPCR was first performed for various proapoptotic and anti-apoptotic genes (Figure 7). In all cell lines, BTZ or combinations of class IIa HDACIs and BTZ affect gene expression estimated by PCR (with few exceptions, such as Bcl-2 in HL-60). HL-60 cells show a marked increase in *BAD* expression (1.3–1.4 log units) upon BTZ treatment (alone or in combination with class IIa HDACI), whereas this increase is much lower (only up to 0.198 log units) in the other two cell lines. In contrast, *p21* is strongly upregulated in K562 and MONO-MAC-6 upon BTZ single or combination treatment but downregulated in HL-60. However, *p21* is also upregulated in HL-60 upon treatment with class IIa HDACI without BTZ. Expression changes of *survivin* are heterogenous. Decreased expression of *survivin* is observed with the combination of BTZ and YAK540 in HL-60, and in MONO-MAC-6 with the combination of BTZ with TMP269 or YAK540. In contrast, expression of *survivin* increased in K562 upon treatment with BTZ alone or in combination with the class IIa HDACIs. Treatment-induced expression changes were also heterogenous for the antiapoptotic proteins Bcl-2 and Bcl-xl. In HL-60 cells, class IIa HDACIs, BTZ and combinations thereof increased expression of *Bcl-2* whereas expression of *Bcl-xl* was decreased, particularly by BTZ and combinations of BTZ and class IIa HDACIs. In K562 cells, bortezomib-containing treatments decreased expression of both, *Bcl-2* and *Bcl-xl*. In MONO-MAC-6, treatments increased *Bcl-xl* and decreased *Bcl-2* expression. Expression of *HDAC 4* was not markedly influenced by any treatment, showing no counter regulation by treatment with class IIa HDACI. Treatment-induced gene expression changes for *PTEN*, *BIM*, *BAK*, *APAF-1*, *MCL-1*, *XIAP*, *p53*, *FOXO1*, and *FOXO3a* were heterogeneous in the 3 cell lines. In summary, PCR data show that treatment with BTZ and class IIa HDACI and combinations thereof have a strong influence on the expression of pro- and antiapoptotic genes, however with no clear tendency towards pro- or anti-apoptosis.

Thus, western blot was used to trace the expression of key proteins in apoptosis, survival, or cell cycle, and to monitor treatment-dependent changes of HDAC 4 expression, induction of PARP cleavage and H2AX phosphorylation (γ-H2AX) (Figure 8).

There is no significant change in HDAC 4 expression upon treatment with class IIa HDACIs, suggesting there is no counter-regulation of HDAC 4 activity upon inhibition by class IIa HDACIs. PARP cleavage can be observed in all cell lines, sometimes more pronounced after 24 h sometimes more pronounced after 48 h, and to a much lesser extent in K562 cells (Figure 8c,d). HL-60 cells show increased PARP cleavage upon combination of YAK540 or TMP269 with BTZ compared to BTZ alone after 24 h (Figure 7a). After 48 h, PARP cleavage was observed under all conditions (Figure 7b). K562 cells show faint PARP cleavage bands after BTZ and BTZ/class IIa HDACI treatment after 48 h (Figure 7d). In MONO-MAC-6 cells after 48 h, PARP cleavage is clearly stronger after combination treatment YAK540 plus BTZ compared to BTZ alone. Interestingly, TMP269 plus BTZ demonstrates no stronger band than BTZ alone (Figure 7f). Phosphorylation of H2AX (γ-H2AX) indicating DNA double strand breaks was only weakly detectable in HL-60 and MONO-MAC-6 cells after 48 h in treatments containing BTZ (Figure 7b,f). In the K562 cells, γ-H2AX was not observed. Proapoptotic proteins BAD and APAF-1 are mostly weakly expressed in all three cell lines with basically no difference between the various treatments. Survivin expression was increased by BTZ in K562 cells (Figure 7c,d) and slightly increased by class IIa HDACIs in HL-60 and MONO-MAC-6 (Figure 7a,b,f). Bcl-2 expression was only observed in MONO-MAC-6 cells und similar under all treatment conditions (Figure 7e,f).

Most prominent changes were observed in the expression of p21 after 24 h treatment (Figure 7a,c,e). BTZ induced a small increase in p21 expression which was markedly enhanced by class IIa HDACIs. Interestingly, in HL-60 and K-562 cells, only YAK540 but not TMP269 increased p21 expression compared to BTZ.

### 2.6. Selectivity of Class IIa HDACI in Combination with BTZ for Leukemia vs. Non-cancer Cell

IC_50_ values of YAK540, TMP269, and BTZ for the non-cancer cell line HEK293 are displayed in Table 4. Corresponding concentration-response-curves are presented in Appendix A.

Then, the combination of class IIa HDACIs and BTZ was tested at HEK293 and for comparison at HL-60 cells. Results are displayed in Figure 9.

Equitoxic concentrations of TMP269 and YAK540 were used based on the IC_50_ values at HEK293 (Table 4). Since the IC50 of YAK540 is > 100 µM, the IC50 of YAK540 is at least 3–4-fold higher than the IC50 of TMP269 which was estimated as 32.8 µM (Table 4). Thus, concentrations used were: 2.5 and 5 µM for TMP269 and 10 and 20 µM for YAK540. The combination of class IIa HDACIs with BTZ was significantly less toxic in the non-cancer cell line HEK293 than in HL-60 for all concentrations tested. As an example, the cytotoxicity of 12 nM BTZ plus 10 µM YAK540 was 44.7% in HEK293 compared to 84.7% in HL-60. Thus, roughly a 2-fold selectivity for leukemia versus HEK293 was observed for the combination treatment.

## 3. Discussion

Sufficient treatment of AML is still an unmet medical need. Combination trials of BTZ with the standard therapy consisting of cytarabine, daunorubicin and etoposide in newly diagnosed pediatric AML patients showed the greatest beneficial effect in a patientsubgroup with reduced phosphorylation of heat shock factor 1 at serine^326^ (low HSF1-pSer^326^) as well as low expression of RelA-pSer^536^. A smaller beneficial effect was demonstrated in subgroups with reduced expression of RelA-pSer^536^ only or reduced expression of HSF1-pSer^326^ only [28,29]. In a subsequent work, the findings regarding HSF1-pSer^326^ were confirmed using in vitro data [29]. No improvement in overall survival and event-free survival was detected considering the entire cohort of the study indicating that innovative strategies are urgently needed for all leukemias, especially AML patients. Novel strategies have included HDACI as combination partners of various cytostatic agents, among them proteasome inhibitors such as BTZ [30]. Whereas so far mainly pan-HDACI such as vorinostat have been used as combination partners, class IIa HDACIs have less been investigated [30]. BTZ is approved for the treatment of multiple myeloma, its use is however limited by severe side effects [5,8,9]. Therefore, strategies to increase the efficacy of proteasome inhibitors and thus to reduce their required dose and consequently their toxic side effects are highly desirable. Switching to ixazomib could reduce toxicity through an improved safety profile, however ixazomib is significantly less potent than BTZ, thus reducing the efficacy of the therapy [10,11,12,13]. Therefore, we dedicated our study to the combination of HDACI with BTZ to investigate whether a synergistic effect would allow a dose reduction of BTZ maintaining the efficacy against leukemias but reducing toxic side effects. Since pan-HDACI and class I HDACI also have severe side effects, we focused in this paper on class IIa HDACI known for their reduced toxicity compared to pan-HDACI [20]. Previous studies have investigated carfilzomib and TMP269, one of the first TFMO-based class IIa HDACI (Figure 1) [5,14,21]. TMP269 has however the problem of limited water solubility [17,31]. We have thus developed YAK540, an improved TFMO-based class IIa HDACI, accessible with a simple straightforward synthesis (Figure 1 and Figure 2) and improved water solubility compared to TMP269. We then compared the anti-leukemia activity of YAK540 in combination with BTZ in comparison to TMP269. Most interestingly, YAK540 has superior effects over TMP269 although inhibition data at recombinant HDAC enzymes are comparable for these two compounds: IC_50_ at HDAC4 is 114 nM for YAK540 and 157 nM for TMP269 (Table 1). Superior effects of YAK540 include an almost 200-fold higher cellular HDAC activity compared to TMP269 and a high selectivity for class IIa HDAC enzymes over class I/IIb enzymes (108fold, Figure 2). Furthermore, YAK540 is 4–7.7 fold less cytotoxic at leukemia and non-cancer cells (Table 2, Table 4) but still increases caspase activation and apoptosis (measured as subG1 fraction) in a synergistic manner at basically the same concentration as used for TMP269 (5 µM TMP269, 5.7 µM YAK540; Figure 4 and Figure 5). YAK540 seems more efficacious than TMP269 as YAK540 is synergistic even at low BTZ concentrations (3.8 nM) at which TMP269 shows no over-additive effects (Figure 4 and Figure 5). The toxicity mechanism of caspase3/7-mediated apoptosis induction is in accordance to literature data where co-incubation of carfilzomib and TMP269 was studied [14].

Chou-Talalay analysis, the gold standard of synergy analysis, was performed in 4 leukemia cell lines using MTT assay. YAK540 and TMP269 were highly synergistic as noticed by CI values below 0.5 (Table 3) and graphical display as blue surface using the Bliss model in Figure 6. Synergy was stronger in 2 of the 4 cell lines (HL-60, MONO-MAC-6; Figure 6a,b,e,f) for YAK540 than for TMP269. Strong synergy for YAK540 was already detected at 10 µM, a concentration that still retains class IIa HDAC preference as seen in enzyme assays and cellular HDAC assays (Table 1, Figure 2). Considering the clearly lower cellular toxicity of YAK540 compared to TMP269 (Table 2 and Table 4), YAK540 treatment could result in fewer side effects than would be expected from the more toxic TMP269.

Toxicity of class IIa HDACI is rather low [20]. YAK540 showed IC_50_ values > 100 µM (except MONO-MAC-6, where an IC_50_ of 91.4 µM was estimated) in leukemia and non-cancer HEK293 cells (Table 2 and Table 4) and is thus clearly less toxic than TMP269. Still, TMP269 was 1.4–2.5 less toxic in HEK293 cells than in the leukemia cells (Table 2 and Table 4), confirming generally low toxicity of class IIa HDACIs. Further, the combination of class IIa HDACI with BTZ was clearly less toxic in non-cancer HEK293 cells than in HL-60 as proven over a wide concentration range (Figure 9) although IC_50_ values of BTZ were similar in leukemia and HEK293 cells. YAK540 is thus clearly advantageous due to lower toxicity than TMP269 and selective toxicity in combination with BTZ for leukemia over non-cancer cells.

A further advantage of YAK540 is the induction of p21 in combination with BTZ compared to the combination of TMP269 and BTZ (Figure 8, except MONO-MAC-6 cells). Our results also supported an advantage for YAK540 over TMP269 due to stronger apoptosis and caspase activation (Figure 4 and Figure 5) which may be based on the detected increased p21 induction upon YAK540/BTZ compared to TMP269/BTZ leading to cell cycle arrest and subsequent apoptosis. Data on PARP cleavage (Figure 8) confirmed these results.

A similar effect on p21 expression has been described for the pan-HDAC inhibitor vorinostat [32]. Expression of p21 was similar in our PCR and western blot data. For other genes/proteins, such correlation between mRNA and protein data could not be confirmed. Still, class IIa HDACI treatment leads to major PCR-detectable changes in expression of pro- and antiapoptotic genes (Figure 7 and Figure 8). Still, p21 seems to be the key regulator for the synergistic effect of YAK540 and BTZ in caspase-mediated apoptosis.

In summary, we have introduced the improved class IIa HDACI YAK540 with advantages over the first TFMO-based class IIa HDACI TMP269. Improved activity (synergy) of YAK540 and selective toxicity for leukemia versus non-cancer cells may allow to reduce the therapeutic dose of BTZ in combination with YAK540 (class IIa HDACI) retaining or yet improving therapeutic efficacy while reducing severe patient-associated toxicity of BTZ. Clinical data have shown a c_max_ value of 580 nM for BTZ after intravenous administration [33]. Regarding IC_50_ values of BTZ estimated in this study (up to 21 nM, Table 2), a dose-reduction of BTZ while maintaining therapeutic efficacy through combination with low toxic YAK540 (class IIa HDACI) is a hypothesis worth being tested in further studies, including animal and clinical studies, to improve multiple myeloma, leukemia and especially AML treatment. Our study reveals synergism between class IIa HDACI (here YAK540) and proteasome inhibitors (here BTZ). This has the potential to improve future treatment of multiple myeloma and/or leukemias like AML using proteasome inhibitors in combination with novel low toxic class IIa HDACI (e.g., YAK540) whereas today the use of proteasome inhibitors often has to be stopped due to severe side effects. Future animal and later clinical studies are needed to investigate the therapeutic benefit of class IIa HDACI in combination with proteasome inhibitors, clinical synergy, and possible dose reduction of proteasome inhibitors due to synergy to reduce severe side effects.

## 4. Materials and Methods

### 4.1. Materials

The culture medium RPMI 1640 and DMEM, Penicillin/streptomycin (pen/strep) (10,000 U/mL; 10 mg/mL) and trypsin-EDTA (0.05% Trypsin, 0.02% EDTA in Phosphate Buffer Saline) were related from PAN-Biotech (Aidenbach, Germany). Bortezomib and TMP269 were ordered from AdooQ (AdooQ Bioscience, Irvine, California, USA). Both compounds were dissolved in DMSO at 10 mM. Further dilutions were made with the appropriate culture medium. A maximum concentration of 1% DMSO was used in all experiments.

### 4.2. Cell Lines and Cell culture

The human promyeloblast leukemia cell line HL-60, the human lymphoblast leukemia Cell line K562 and the monocytic leukemia Cell line MONO-MAC-6 and the human monocytic leukemia Cell line THP-1 were kindly provided by Helmut Hanenberg, Department of Otorhinolaryngology and Head/Neck Surgery, Heinrich Heine University, 40,225 Düsseldorf, Germany. All leukemia cell lines were cultured with RPMI 1640 spiked with 120 IU/mL penicillin and 120 µg/mL streptomycin and 10% heat inactivated fetal calf serum at 37 °C and 5% CO_2_. The non-cancer cell line HEK293 was acquired by DSMZ (Braunschweig, Germany). The cells were cultured with DMEM mixed with 120 IU/mL penicillin and 120 µg/mL streptomycin and 10% heat inactivated fetal calf serum at 37 °C and 5% CO_2_.

### 4.3. MTT Cell Viability Assay

Cell viability during treatment with BTZ and the class IIa inhibitors YAK540 and TMP269, respectively, and concentration-response curves were determined by MTT assay with minor modifications as previously described [34,35]. Briefly, cells were incubated with the compounds for 72 h. The following cell numbers were plated: HL-60 25.000 c/w, K562 35.000 c/w, MONO-MAC-6 30.000 c/w, and THP-1 15.000 c/w. After 72 h, 25 µL MTT (3-(4,5-dimethylthiazol-2-yl)-2,5-diphenyltetrazolium bromide, Serva, Heidelberg, Germany) solution (5 mg/mL in phosphate buffer saline) was added. The formed formazan was dissolved by pipetting up and down after addition of 150 µL isopropanol/concentrated hydrochloric acid (50 mL/165 µL). Subsequently, the absorbance option was measured at 595 nm and 690 nm on the microplate reader ThermoFisher Multiskan FC Microplate Photometer (Thermo Scientific, Wesel, Germany).

### 4.4. Enzyme HDAC Inhibition Assay

The human recombinant enzymes were ordered from Reaction Biology Corp (Malvern, PA, USA). To determine the inhibitory activity of the enzymes HDAC 2 (cat. no. KDA-21-277), HDAC 4 (cat. nr. KDA-21-279), HDAC 6 (cat. no. KDA-21-213), and HDAC 8 (cat. no. KDA-21-481), 20 ng each of HDAC 2 and 8, 17.5 ng of HDAC 6, and 2 ng of HDAC 4 were used per well. After pipetting 10 µL of increasing concentrations of YAK540 and TMP269 to a 96-well plate (Corning, Kaiserslautern, Germany), recombinant enzymes diluted in assay buffer (50 mM Tris-HCl, pH 8.0, 137 mM KCl, 1 mM MgCl_2_, and 1 mg/mL BSA) were added. After five minutes of incubation, the reaction was started with 10 µL of a 30 µM solution of Boc-Lys(Ac)-AMC (Bachem, Bubendorf, Switzerland) for HDAC 2 or 15 µM solution of Boc-Lys(Ac)-AMC for HDAC 6 or a 10 µM solution of Boc-Lys-(Tfa)-AMC (Bachem, Bubendorf, Switzerland) for HDAC 4 or a 6 µM solution of Boc-Lys-(Tfa)-AMC for HDAC 8. After 90 min, the reaction was stopped by adding 100 µL of stop buffer (16 mg/mL trypsin and 4 µM vorinostat for HDAC 2, 2 µM CHDI00390576 for HDAC 4, 4 µM tubastatin A for HDAC 6 and 4 µM panobinostat for HDAC 8 in 50 mM Tris-HCl, pH 8.0 and 100 mM NaCl). After an additional 15 min, fluorescence intensity was measured at an excitation wavelength of 355 nm and an emission wavelength of 460 nm in the NOVOstar microplate reader (BMG LabTech, Offenburg, Germany).

### 4.5. Whole-Cell HDAC Inhibition Assay

The cellular HDAC assay is based on the publications by Heltweg and Jung [36], Ciossek et al. [37] and Bonfils et al. [38] Minor modifications were still made as described in [39]. In 96-well plates (Corning, Kaiserslautern, Germany), 25,000 c/w THP-1 were seeded in 90 µL culture medium. After 24 h, increasing concentrations of YAK540 and TMP269 were added and incubated for an additional 18 h. Then, the reaction was started by adding 10 µL of 3 mM Boc-Lys(Ac)-AMC or 1 mM Boc-Lys(Tfa)-AMC (Bachem, Bubendorf, Switzerland) to achieve a final concentration of 0.3 mM Boc-Lys(Ac)-AMC or 0.1 mM Boc-Lys(Tfa)-AMC, respectively. After three hours of incubation under culture conditions, 100 µL/well stop buffer (25 mM Tris-HCl, pH 8.0, 127 mM KCl, 1 mM MgCl2, 1% NP40, 2 mg/mL trypsin, 10 µM panobinostat) was added. After another three hours under culture conditions, fluorescence intensity was measured at an excitation wavelength of 320 nm and an emission wavelength of 520 nm in the NOVOstar microplate reader (BMG LabTech, Offenburg, Germany).

### 4.6. Measurement of Apoptotic Nuclei

THP-1 cells were seeded at a density of 10.000 c/w in 24-well plates (Sarstedt, Nürnbrecht, Germany). Cells were treated with BTZ and/or HDACI for the indicated time. Supernatant was removed after a centrifugation step and the cells were lysed in 500 µL hypotonic lysis buffer (0.1% TritonX-100, 100 µg/mL propidium iodide) at 4 °C in the dark overnight. Triton X-100 was obtained from AppliChem (Darmstadt, Germany). Propidium iodide was delivered by Santa Cruz Biotechnology (Heidelberg, Germany). The percentage of apoptotic nuclei with DNA content in sub-G1 was analyzed by flow cytometry using the CyFlow instrument (Partec, Norderstedt, Germany).

### 4.7. Activation of Caspase 3/7

Compound-induced activation of caspase 3/7 was analyzed using the CellEvent Caspase 3/7 green detection reagent (Thermo Scientific, Wesel, Germany) according to the manufacturer’s instructions. Briefly, THP-1 cells were seeded in 96-well-plates (Corning, Kaiserslautern, Germany) at a density of 10.000 c/w. Cells were treated with BTZ and/or HDACI for the indicated time. Then, a solution of CellEvent Caspase 3/7 green detection reagent in cell culture medium supplemented with 5% heat inactivated FCS for a final assay concentration of 10 µM was added. Cells were incubated for 30 min at 37 °C under cell culture conditions before imaging by using the Thermo Fischer ArrayScan XTI high content screening (HCS) system with a 10 × magnification (Thermo Scientific, Wesel, Germany). Hoechst 33,342 was used for nuclei staining at a concentration of 10 mg/mL in water (Sigma-Aldrich, Steinheim, Germany).

### 4.8. Immunoblotting

Protein extraction and Western blot analysis were performed as previously described with minor modifications [34]. Cells were incubated for 24 h or 48 h with the concentrations listed. Cells were centrifuged after the incubation period and the medium was aspirated. Cells were washed with phosphate buffer saline and centrifuged again. After re-aspiration, RIPA buffer (50 mM Tris-HCl pH 7.4, 1% NP-40, 0.5% sodium deoxycholate, 0.1% SDS, 150 mM sodium chloride, 2 mM EDTA, spiked with protease and phosphatase inhibitor (Pierce protease and phosphatase inhibitor mini tablets, Thermo Scientific, Wesel, Germany)) was added and the pellet was lysed for 30 min at 4 °C. Then the protein preparations were boiled at 95 °C with 2x Laemmli buffer containing β-mercaptoethanol for 3 min. Pierce BCA protein assay was used to determine the total protein content in the protein preparations (Thermo Scientific, Rockford, IL, USA). Equal amounts of protein (40 µg, except for mentioned variations) were separated by SDS-PAGE and transferred to a polyvinylidene fluoride membrane (Merck Millipore, Darmstadt, Germany). PageRuler Prestained Ladder, 10 to 180 kDa (Thermo Scientific, Wesel, Germany) was used as a size comparison marker. Blots were incubated with the primary antibodies (Appendix A). The secondary antibodies conjugated to HRP used in the Western blot were supplied by R&D Systems (Minneapolis, MN, USA). Proteins of interest were visualized using Luminol reagent (Santa Cruz Biotechnology, Heidelberg, Germany) on Intas Imager (Intas, Göttingen, Germany).

### 4.9. RT-PCR

Cells were incubated with the intentional concentrations of BTZ, TMP269, and YAK540 for 48 h. RNA was then isolated using the RNeasy Mini Kit (Qiagen, Hilden, Germany). Subsequently, the RNA was transcribed into cDNA using the High-Capacity cDNA Reverse Transcription Kit (Thermo Scientific, Wesel, Germany). For RT-PCR, GoTaq qPCR Master Mix (Promega, Fitchburg, WI, USA) and a CFX96 Real-Time System (BIO-RAD, Hercules, CA, USA) were used. The relative changes in gene expression were normalized to the control genes *GUSB* (beta-glucuronidase), *TBP* (TATA binding protein), and *HPRT1* (hypoxanthine-guanine phosphoribosyltransferase) by Vandesompele method [23]. Primers were designed using Primer-BLAST (NIH, Bethesda, Maryland, USA) and had an efficiency between 80% and 115% (Table 5). Efficiency was determined using an isolated template of HeLa cells.

## 5. Conclusions

We have developed the improved TFMO-based class IIa selective HDACI YAK540 and tested this inhibitor in combination with BTZ for synergy in comparison to the first generation TFMO class IIa HDACI TMP269 to investigate the basis for improved treatment of leukemia, particularly AML. Advantages of YAK540 over TMP269 are simple, straight-forward synthesis; improved solubility; 4-fold less toxic in non-cancer HEK293 cells; 200-fold stronger cellular HDAC activity but yet similar HDAC enzyme activity and class IIa selectivity; more efficacious in inducing caspase-mediated apoptosis; stronger synergy in 2 of 4 examined leukemia cell lines; stronger induction of p21; similar cytotoxic selectivity for leukemia versus HEK293 cells in combination with BTZ. The strong synergy of YAK540 with BTZ may allow reducing currently clinically used doses of BTZ limiting the use of proteasome inhibitors. Synergistic combination treatment of a class IIa HDACI and a proteasome inhibitor would allow reducing severe toxicity of proteasome inhibitors but retaining clinical anticancer efficacy. Based on our promising results, we suggest further investigation of YAK540 in combination with proteasome inhibitors in leukemia, particularly AML.

## Data Availability

The data presented in this study are available in this article and the corresponding Appendix A.

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
