# Peer review of "The Novel Class IIa Selective Histone Deacetylase Inhibitor YAK540 Is Synergistic with Bortezomib in Leukemia Cell Lines"

_ijms, 2022, doi:10.3390/ijms232113398_

Round 1
Reviewer 1 Report
The authors developed the improved TFMO-based class IIa selective HDACI YAK540 and tested this inhibitor in combination with BTZ for synergy in comparison to the first generation TFMO class IIa HDACI TMP269. They showed that the combination of YAK540 and BTZ showed generally increased proapoptotic gene expression, increased p21 expression, and synergistic, caspase 3/7-mediated apoptosis. Notably, the cytotoxicity of YAK540 is much lower than that of pan-HDACIs. Further, combinations of YAK540 and BTZ are clearly less toxic in non-cancer HEK293 compared to HL-60 leukemia cells.
This strategy is important for leukemia treatment and this manuscript was well written.
Please indicate the clinical implication of this study for more effective treatment of leukemias as well as multiple myeloma in the Discussion section.
Reviewer 2 Report
In this manuscript, “The novel Class IIa Selective Histone Deacetylase Inhibitor YAK540 is synergistic with Bortezomib in Leukemia Cell Lines” Lukas M. Bollmann et. Al. demonstrated the synergistic combination of class IIa selective HDACIs such as YAK540 and proteasome inhibitors represents a promising approach against leukemias to increase the anticancer effect and to reduce the general toxicity of HDACIs. Here authors proposed and demonstrated experimentally that YAK540 is more selective Class IIa HDAC inhibitor with significant lower cytotoxicity compared to TMP269. Authors have done excellent job in compiling the big amount of data in a short and more representative way. The data well supports the novelty of YAK 540 to be used in combination with proteosome inhibitor without affecting the normal cells.
The comments and suggestions for this manuscript are as follows-
1. In figure 1 the green color text has less visibility. Therefore, choosing other dark color can improve the visibility.
2. Authors have shown the caspase 3/7 activation using the Cell Event 534 Caspase-3/7 green detection reagent. Showing western blot results for this activation would be more convincing.
3. In figure 8, western blot results for Survivin and APAF-1 is very faint.
4. Authors should include following recent publications in their manuscript.
https://pubmed.ncbi.nlm.nih.gov/34719869/
https://pubmed.ncbi.nlm.nih.gov/32959058/
https://pubmed.ncbi.nlm.nih.gov/33802801/
